# Airspace Enlargement with Fibrosis in a Young Heavy Smoker Mimicking Diffuse Cystic Lung Disease

**DOI:** 10.3390/medicina58111648

**Published:** 2022-11-15

**Authors:** Kyungsoo Bae, Hyo Jung An, Kyung Nyeo Jeon

**Affiliations:** 1Department of Radiology, Institute of Health Sciences, Gyeongsang National University School of Medicine, Jinju 52727, Republic of Korea; 2Department of Radiology, Gyeongsang National University Changwon Hospital, Changwon 51472, Republic of Korea; 3Department of Pathology, Gyeongsang National University School of Medicine, Jinju 52727, Republic of Korea; 4Department of Pathology, Gyeongsang National University Changwon Hospital, Changwon 51472, Republic of Korea

**Keywords:** lung cysts, computed tomography, airspace enlargement with fibrosis, smoking, pulmonary Langerhans cell histiocytosis, lymphangioleiomyomatosis, emphysema

## Abstract

The widespread use of computed tomography (CT) has led to the increased recognition of cystic lung lesions. Multiple pulmonary cysts can be observed in heterogeneous disorders called diffuse cystic lung diseases (DCLDs), which include pulmonary Langerhans cell histiocytosis, lymphangioleiomyomatosis, lymphocytic interstitial pneumonia, and Birt–Hogg–Dubé syndrome. Recently, airspace enlargement with fibrosis (AEF) has been recognized as an entity on the spectrum of smoking-related lung diseases. We report a young male heavy smoker presenting diffuse pulmonary cysts on chest CT with suspected DCLD. However, histopathological examination of the surgical biopsy specimen revealed dilated emphysematous cysts with prominent mural fibrosis, consistent with AEF.

## 1. Introduction

Owing to the widespread use of computed tomography (CT), cystic lung lesions are frequently detected on chest CT scans. The characteristic features of diffuse cystic lung diseases (DCLDs) such as pulmonary Langerhans cell histiocytosis (PLCH) and lymphangioleiomyomatosis (LAM) are well established [1]. However, in practice, making a specific diagnosis is difficult due to non-specific clinical features or confounding radiologic findings. We present a young male heavy smoker with multiple lung cysts mimicking DCLD. However, a final diagnosis of airspace enlargement with fibrosis (AEF) was made through a histopathological examination of surgical biopsy specimens.

## 2. Case Report

This study was approved by the Institutional Review Board (IRB) of our hospital. The requirement of informed consent was waived by the IRB. A 34-year-old male patient presented with intermittent cough and chest tightness for several years. The patient had a history of smoking two packs of cigarettes per day for 20 years (a 40-pack-year smoking history). The patient’s past medical history was unremarkable. He has worked as an office worker. A chest X-ray was unremarkable. Laboratory findings were also unremarkable, except for an increased serum cholesterol level at 238 mg/dL. A pulmonary function test showed a forced expiratory volume in 1 s/forced vital capacity (FEV1/FVC) of 92% pred, an FEV1 of 92% pred, and a maximal forced expiratory flow (MFEF) 75/25 of 60% pred.

A chest CT scan revealed multiple pulmonary cysts of variable sizes and shapes (Figure 1a–d). The intervening lungs were clear. Cysts were mainly distributed in the peribronchial areas of upper and middle lung zones. Both basal lungs, including costophrenic angles, were relatively spared. Some small round cysts showed rarely perceptive walls, consistent with emphysema. However, most cysts showed irregular, branched, or bizarre shapes with well-defined walls. A few micronodules were observed in the periphery of both upper lobes. Therefore, PLCH of the predominantly cystic type was suspected. A video-assisted thoracoscopic wedge biopsy was performed from the right upper lobe.

Histopathologic examination revealed multiple cystically dilated air spaces of variable sizes (Figure 1e). Cysts corresponding to irregular, branching, or bizarre cysts on chest CT showed thick fibrous walls and a connection with the distal bronchioles (Figure 1f). Some small thin-walled cysts corresponded to emphysematous cysts without fibrosis in their walls. Tiny focal nodular areas with stellate cell aggregation were noted. However, residual Langerhans cell aggregation was absent on morphological assessment and immunohistochemical staining for S100 and CD1a (Figure 1g). Based on pathologic findings, a final diagnosis of AEF was made. Subsequent evaluations conducted after the biopsy did not reveal any other co-morbidities such as autoimmune diseases, genetic disorders, or infectious conditions potentially causing cystic lung disease. The patient’s serum alpha1-anti-trypsin level was within a normal range (140 mg/dL). Lung cysts did not show interval changes on follow-up CT scans for three years. The patient has been stable without respiratory symptoms since quitting smoking cigarettes.

## 3. Discussion

Smoking-related chronic lung diseases comprise heterogenous disorders such as respiratory bronchiolitis–interstitial lung disease, desquamative interstitial pneumonia, usual interstitial pneumonia, PLCH, and centrilobular emphysema. Of these, PLCH and centrilobular emphysema can manifest as multiple cysts in the upper to middle lung zones in young smokers. Emphysema appears as centrilobular lucent areas without distinct walls due to the absence of mural fibrosis. The centrilobular artery can be observed in the center of the lucent area [2]. PLCH is the prototypical DCLD. It has characteristic CT findings such as a mix of nodules, cavitary nodules, and cysts in the upper to middle lung zones. As PLCH disease progresses, the cysts grow larger and in bizarre shapes, such as irregular, bi-lobed, cloverleaf, and branched, due to the fusion of adjacent cysts [3]. Advanced burnt-out PLCH may mimic centrilobular emphysema, further confounding CT findings [3].

In this study, a young individual who smoked heavily showed multiple cysts of variable sizes and shapes in the upper and middle lung zones. Some small cysts were consistent with centrilobular emphysema without definable walls. However, most cysts showed obvious walls and lobulated, branched, or bizarre shapes. On pathologic examination, the cysts were different from emphysema and cystic PLCH. Recently, emphysematous cysts with fibrotic walls and structural remodeling have been recognized as frequent findings in the background lung of specimens resected for lung cancer [4]. Those cysts are termed as AEF and are categorized as a separate entity in smoking-related interstitial pneumonia according to a recent update of idiopathic interstitial pneumonias [4,5]. However, radiologic findings of AEF are not yet widely known. Watanabe et al. reported that multiple, thin-walled cysts are among high resolution CT features of AEF [6].

Cigarette smoke can lead to alveolar wall fibrosis that can increase with time and intensity of exposure [7]. AEF frequently accompanies centrilobular emphysema. Yamada et al. [8] suggested a common mechanism of tissue damage between AEF and centrilobular emphysema in their histological and histochemical analysis of lung specimens from 39 patients. Gupta et al. [9] reported pathologic and CT findings of four patients after exposure to cigarette smoke with multiple lung cysts similar to ours. They concluded that cystic change might be an unusual manifestation of smoking-related small airway injury, although factors that might favor cyst formation rather than a more common emphysema are unclear [8]. Most patients with AEF have stable disease and a good survival time [7]. Progressively worsening respiratory symptoms in patients with AEF might be due to concurrent chronic interstitial pneumonia such as idiopathic interstitial fibrosis [7].

In summary, we reported a young male presenting multiple well-defined irregular pulmonary cysts mimicking DCLD on chest CT. He was finally diagnosed as AEF. Thus, cigarette smoking-related small airway damage presenting AEF should be considered as an alternative cause of multiple lung cysts, particularly in smokers.

## Figures and Tables

**Figure 1 medicina-58-01648-f001:**
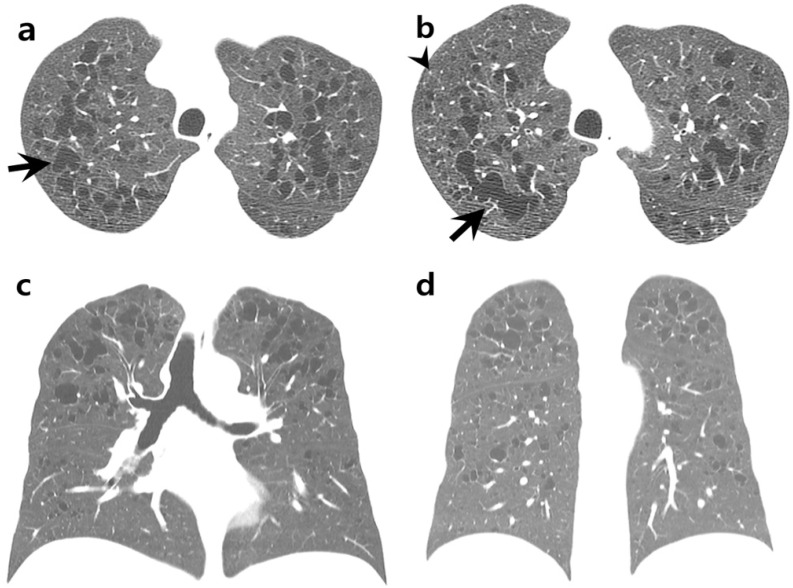
A 34-year-old male with intermittent cough and chest tightness. (**a**,**b**) Axial chest CT images with lung window setting showing cysts of variable sizes in both upper lungs. Intervening lungs are clear. Most cysts show prominent walls with branched, lobulated, or bizarre shapes (arrows). A few micronodules are observed in the periphery of the right upper lobe (arrowhead in (**b**)). (**c**,**d**) Coronal chest CT images showing distribution of cysts, predominantly in the upper and middle lung zones. Both basal lungs and costophrenic angles are relatively spared. (**e**) Resected lung tissue showing cystically dilated space with hyalinized fibrosis (arrows) (×10, Hematoxylin and Eosin staining). Two small nodular areas with stellate cell aggregation are marked with circles. (**f**) Pigmented macrophages with finely granular cytoplasm are accumulated in distal and peribronchiolar airspace. There is no definite diffuse interstitial thickening or inflammation (×400, Hematoxylin and Eosin staining). (**g**) Magnification view of an irregular cyst (star in Figure 1e) showing thickened cystic wall with fibrotic change (×100, Hematoxylin and Eosin staining). (**h**) Magnification view of a small nodular area marked in Figure 1e showing stellate cell aggregation (×400, Hematoxylin and Eosin staining). (**i**,**j**) Only a few stellate cells showed positive expression for CD1a (**i**: ×200, **j**: ×400). Note presence of anthracotic pigmentation (arrow in Figure 1i) nearby.

## Data Availability

Data are contained within the article. No new data were created or analyzed in this study.

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
