# Peer review of "Airspace Enlargement with Fibrosis in a Young Heavy Smoker Mimicking Diffuse Cystic Lung Disease"

_medicina, 2022, doi:10.3390/medicina58111648_

Round 1

Reviewer 1 Report

I congratulate with the authors, the work is well structured, interesting and scientifically valid. I have only a few minor considerations.

Minor questions:

1)      The authors in section 2 describe the patient as a heavy smoker. I would suggest, however, to include (if possible) also how many years the patient has started smoking (if the data is available).

2)      On page 1 in the case report section, where the authors list the salient characteristics of the patient described in this study I would suggest to the authors to specify that the patient is not suffering from other pathologies that may affect the lung (such as bronchial asthma, cystic fibrosis aspergillosis other infections or other condictions) it may seem obvious but in my opinion it would be better to specify it, given that in the literature the syndromes of diffuse cystic lung diseases can have various and heterogeneous causes.

3)      Finally, a personal curiosity. since the importance of pro-inflammatory cytokines such as IL-1 and TNF alpha causing lung inflammation characterized by infiltration of neutrophils and macrophages, with distal airspace enlargement, disruption of elastin fibers in alveolar and fibrousis my curiosity is, Did the authors try to analyze whether on bronchoaspirate / broncholwash samples (if available) or on biopsy histological sections (with immu-histochemical technique) they could verify an increased expression of proinflammatory cytokines and chemokines? has the data been investigated ? if it yes would be interesting to add this data as well.

Author Response

I congratulate with the authors, the work is well structured, interesting and scientifically valid. I have only a few minor considerations.

Minor questions:

1)  The authors in section 2 describe the patient as a heavy smoker. I would suggest, however, to include (if possible) also how many years the patient has started smoking (if the data is available).

A) We thank the reviewer for pointing this out. We have edited the sentence regarding the patient’s smoking history as shown below:

The patient was a heavy smoker with a history of 40 pack-years of smoking →

The patient had a history of smoking two packs of cigarettes per day for 20 years (a 40-pack-year smoking history).

2)   On page 1 in the case report section, where the authors list the salient characteristics of the patient described in this study I would suggest to the authors to specify that the patient is not suffering from other pathologies that may affect the lung (such as bronchial asthma, cystic fibrosis aspergillosis other infections or other conditions) it may seem obvious but in my opinion it would be better to specify it, given that in the literature the syndromes of diffuse cystic lung diseases can have various and heterogeneous causes.

A) We thank the reviewer for pointing this out and we agree with the reviewer. Therefore, we have provided additional medical history for the patient.

Subsequent evaluation conducted after the biopsy did not reveal any other comorbidities such as autoimmune diseases, genetic disorders, or infectious conditions potentially causing cystic lung disease.

3)  Finally, a personal curiosity. since the importance of pro-inflammatory cytokines such as IL-1 and TNF alpha causing lung inflammation characterized by infiltration of neutrophils and macrophages, with distal airspace enlargement, disruption of elastin fibers in alveolar and fibrousis my curiosity is, Did the authors try to analyze whether on bronchoaspirate / broncholwash samples (if available) or on biopsy histological sections (with immu-histochemical technique) they could verify an increased expression of proinflammatory cytokines and chemokines? has the data been investigated ? if it yes would be interesting to add this data as well.

A) We thank the reviewer for pointing this out and we agree with the reviewer. However, the patient did not undergo bronchoscopy. Immunohistochemical staining of biopsy specimens could not verify the increased expression of cytokines or chemokines.

We thank the reviewer for the valuable comments and suggestions. These comments and suggestions have improved the quality of our manuscript significantly.

Reviewer 2 Report

Kyungsoo Bae and colleagues here present an interesting case of distal airway enlargement with fibrosis in a young heavy smoker. The case is well presented although I have a few major concerns:

1) Clinical data. Clinical depiction of the patient is scarce and needs to be improved. Were alpha1-anti-tripsin levels evaluated? Were occupational exposures ruled out? Please improve the patient clinical data.

2) Histologic data. Histologic images are quite shallow and do not fully support the diagnosis. As an heavy smoker I would expect some extent of RB-ILD/RBF-ILD and  the presence of smoker's macrophages that are completely absent in the images. Lung parenchyma seems to suffer from inadeguate fixation with collapse artifacts, false appearance of interstitial inflammation and diffuse interstitial fibrosis. The areas depicted as fibrous PLCH have to be shown at an higher magnification or they can be interpreted as collapsed parenchyma. A major reworking of the histologic section is need to better support the diagnosis od distal airway enlargement with fibrosis.

Author Response

Kyungsoo Bae and colleagues here present an interesting case of distal airway enlargement with fibrosis in a young heavy smoker. The case is well presented although I have a few major concerns:

1) Clinical data. Clinical depiction of the patient is scarce and needs to be improved. Were alpha1-anti-tripsin levels evaluated? Were occupational exposures ruled out? Please improve the patient clinical data.

A) We thank the reviewer for pointing this out and we agree with the reviewer. Therefore, we have added more information as shown below:.

The patient’s past medical history was unremarkable. He has worked as an office worker.

Subsequent evaluation conducted after the biopsy did not reveal any other co-morbidities such as autoimmune diseases, genetic disorders, or infectious conditions potentially causing cystic lung disease. His serum alpha1-anti-trypsin level was within a normal range (140 mg/dL).

2) Histologic data. Histologic images are quite shallow and do not fully support the diagnosis. As an heavy smoker I would expect some extent of RB-ILD/RBF-ILD and the presence of smoker's macrophages that are completely absent in the images. Lung parenchyma seems to suffer from inadequate fixation with collapse artifacts, false appearance of interstitial inflammation and diffuse interstitial fibrosis. The areas depicted as fibrous PLCH have to be shown at an higher magnification or they can be interpreted as collapsed parenchyma. A major reworking of the histologic section is need to better support the diagnosis of distal airway enlargement with fibrosis.

A) We thank the reviewer for these valuable comments and we agree with the reviewer. Therefore, we have revised the manuscript as shown below:

We have replaced Fig. 1g with one with a higher magnification (Fig. 1h in the revised version) showing a nodular area with stellate cell aggregation, which the reviewer suspected as collapsed parenchyma in the previous version.

We have added Fig. 1f (in the revised version) showing pigmented macrophages accumulated in distal and peribronchiolar airspaces without definite diffuse interstitial thickening or inflammation.

We thank the reviewer for the valuable comments and suggestions. These comments and suggestions have improved the quality of our manuscript significantly.

Round 2

Reviewer 2 Report

The manuscript has been improved although I am not yet fully convinced about the evidence of P-LCH lacking eosinophilic infiltrated as well, please provide CD1a immunohistochemistry to support your diagnostic interpretation.

Figure F much clarifies the picture of RBFILD / SR lung disease which can be related to emphysematous cysts in such a young heavy smoker.

Author Response

Answers to Reviewers’ Comments and Suggestions

(Reviewer 2)

The manuscript has been improved although I am not yet fully convinced about the evidence of P-LCH lacking eosinophilic infiltrated as well, please provide CD1a immunohistochemistry to support your diagnostic interpretation.

Figure F much clarifies the picture of RBFILD / SR lung disease which can be related to emphysematous cysts in such a young heavy smoker.

A) We thank the reviewer for pointing this out and we agree with the reviewer. Therefore, we have added two more figures showing CD1a immunohistochemical staining.

We thank the reviewer for the valuable comments and suggestions. These comments and suggestions have improved the quality of our manuscript significantly.